# Cognitive Skills and DNA Methylation Are Correlating in Healthy and Novice College Students Practicing Preksha Dhyāna Meditation

**DOI:** 10.3390/brainsci13081214

**Published:** 2023-08-17

**Authors:** Bassam Abomoelak, Ray Prather, Samani U. Pragya, Samani C. Pragya, Neelam D. Mehta, Parvin Uddin, Pushya Veeramachaneni, Naina Mehta, Amanda Young, Saumya Kapoor, Devendra Mehta

**Affiliations:** 1Gastrointestinal Translational Laboratory, Arnold Palmer Hospital for Children, Orlando, FL 32806, USA; bassam.abomoelak@orlandohealth.com; 2Pediatric Cardiothoracic Surgery Department, Arnold Palmer Hospital for Children, Orlando, FL 32806, USA; ray.prather@orlandohealth.com; 3Department of Religions and Philosophies, University of London, London WC1H 0XG, UK; upragya108@gmail.com; 4Department of Biostatistics, Robert Stempel College of Public Health and Social Work, Florida International University, Miami, FL 33199, USA; chaitanyapragya97@gmail.com; 5Sidney Kimmel Medical College, Thomas Jefferson University, Philadelphia, PA 19107, USA; neelamehta77@gmail.com; 6College of Arts, Sciences and Education, Florida International University, Miami, FL 33199, USA; puddi001@fiu.edu; 7College of Law, Florida International University, Miami, FL 33199, USA; pushya@fiu.edu; 8Neurodevelopmental Pediatrician, Behavioral and Developmental Center, Orlando Health, Orlando, FL 32805, USA; mehtade@gmail.com; 9Institute for Simulation and Training, University of Central Florida, Orlando, FL 32765, USA; amanda.young@ucf.edu; 10Medical School, University of Central Florida, Orlando, FL 32827, USA; kapoorsaumya@knights.ucf.edu

**Keywords:** Preksha meditation, cognitive skills, DNA methylation, correlation, cellular pathways

## Abstract

The impact of different meditation protocols on human health is explored at the cognitive and cellular levels. Preksha Dhyana meditation has been observed to seemingly affect the cognitive performance, transcriptome, and methylome of healthy and novice participant practitioners. In this study, we performed correlation analyses to investigate the presence of any relationships in the changes in cognitive performance and DNA methylation in a group of college students practicing Preksha Dhyāna (N = 34). Nine factors of cognitive performance were assessed at baseline and 8 weeks postintervention timepoints in the participants. Statistically significant improvements were observed in six of the nine assessments, which were predominantly relating to memory and affect. Using Illumina 850 K microarray technology, 470 differentially methylated sites (DMS) were identified between the two timepoints (baseline and 8 weeks), using a threshold of *p*-value < 0.05 and methylation levels beyond −3% to 3% at every site. Correlation analysis between the changes in performance on each of the nine assessments and every DMS unveiled statistically significant positive and negative relationships at several of these sites. The identified DMS were in proximity of essential genes involved in signaling and other important metabolic processes. Interestingly, we identified a set of sites that can be considered as biomarkers for Preksha meditation improvements at the genome level.

## 1. Introduction

Meditation and yoga practices have evidenced their efficacy in improving the life quality for meditation participants. Such practices are known to reduce anxiety, stress, and depression, and they have proved to be valid clinical intervention tools for the treatment of depression, ADHD, pain management, drug abuse, and addiction [1,2,3,4,5,6]. Preksha Dhyāna meditation (PM) demonstrated performance-enhancing effects on the psycho-emotional assessments and beneficial changes in the gene profiling and DNA methylation of novice college students [7,8,9]. In addition, PM proved its efficacy in reducing the clinical symptoms such as abdominal pain, stress, and vomiting when tested in a cohort of children with functional abdominal pain disorders (FAPDs) [10]. Furthermore, PM was found to affect several pathways when analyzed at the transcriptome and methylome levels [8,9]. At the cellular level, yoga sessions were found to affect global gene expression in peripheral blood mononuclear cells (PBMCs) [11]. Relaxation was also found to induce temporal transcriptome alterations in inflammatory pathways and energy metabolism [12]. Recently, a link between meditation and the immune system, human microbiota, and epigenetics was proposed [13].

Although practices and protocols may vary, the positive effects of meditation on human health are generally attributed to the reduction in stress through self-recognition, purification of emotions, and self-regulation in general [6]. Various researchers have begun investigating the physiological changes associated with the practice of meditation, but the mechanistic pathways responsible for the changes in higher-order functioning (e.g., cognitive processes, affect, behavior, and other states) remain largely unknown [3,14,15,16]. For meditation and yoga practices to transcend the current role of being an auxiliary tool in the treatment of clinical conditions to becoming, itself, actualized as a validated clinical intervention, elucidation of the genetic and cellular changes responsible for those improvements in higher-order functioning must be achieved. Studies correlating the two levels are lacking. These sorts of studies are always hampered by experimental design, sample size, and/or the expense, but preliminary research must start somewhere. In this report, we assess the correlation between improvements in cognitive skill performance and changes in methylation at the genome level in 34 healthy and novice college students participating in an 8-week course of meditation practice.

## 2. Materials and Methods

### 2.1. Participants and Study Design

The study was approved by the Florida International University (FIU) ethics and IRB committees. The study was registered in the U.S. National Library of Medicine (Clinicaltrials.gov). The study ID was NCT03779269. Of the 142 healthy and novice college students enrolled in the study, 34 participants subjected to a combined sound and color meditation condition were included in our analyses. All participants provided formal, written consent to be enrolled in the study and were given ID numbers in accordance with IRB standards. Participants attended three guided PM sessions each week for a total of eight weeks. The meditation protocols, participant assignments, and data collection procedures were described in detail in our previous publications [7,8,9].

### 2.2. Participants and PM Meditation Protocol

The 34 participants included 6 males and 28 females, and their ages ranged from 18 to 24 years old. The total duration of the meditation and yoga session was 60 min divided into several sessions of 11 min each. A basic universal sound of bee-like buzzing (achieved using several deep breaths and exhaling with closed lips for several seconds at a time while focusing on the vibrations in the head) is repeated to achieve intense concentration. Additionally, focused meditation on a green-colored object like a tree is carried out. Participants concentrated on the visualized green image just in front of the forehead and are guided through imagery emanating from the tree. Between focused meditation periods, a version of alternate nostril breathing, anulom vilom, and varying combinations of the following yoga poses are used: Vajrasana, Shashankasana, Ushtrasana, Marjariasana, Padhastasana, Dhanurasana, Trikonasana in two variations, Pawanmuktasana, and Paschimottanasana.

### 2.3. Test Assessments and Data Management

A battery of assessments was administered, and survey data were obtained to evaluate the cognitive performance (namely, memory and attentional capacities) and affective state of participants in this study. The Automated Working Memory Assessment—Short Form (AWMA-S) was used to assess six facets of short-term and working memory including the following: digit recall and digit recall processing; language recall and language recall processing; and spatial recall and spatial recall processing. The AWMA-S is a construct that involves tasks of recalling increasing complex stimuli (e.g., numeric, auditory, and geometric) and has been validated in up to 22 years of age [17]. The Conners Continuous Performance Test Third Edition™ (Conners CPT 3™) was used to obtain measures of inattentiveness, impulsivity, and sustained attention. The test is a series of letters flashing on the screen with a variable time frame between letters but averaging 250 milliseconds. The respondents are asked to press the spacebar after each letter but omitting this for all X letters that show (presented 36 times among the 324 letters presented). Respondent error and reaction time are analyzed to determine the attention-related capacity scores. The Conner’s CPT has been used to assess the impact of mindfulness in young adults with ADHD [18]. Measure of participants’ affective state was finally obtained using the Positive and Negative Affect Score (PANAS). This short questionnaire consists of two 10-item self-report mood scales measuring the distinct dimensions of positive and negative affect [19]. The Positive Affect scale reflects the extent to which a person feels enthusiastic, active, and alert; the Negative Affect scale reflects the experience of unpleasant mood states, such as nervousness, distress, and irritability. The two scales have been shown to be highly internally consistent, uncorrelated, and stable over time. Mahaprana Dhvani is a humming or buzzing sound produced in the practice of PM. Participant adherence to the meditation practice was assessed by measuring the duration of cued Mahaprana Dhvani (referred to as “Buzz” in our analyses) at different points throughout the meditation sessions.

### 2.4. Blood Withdrawal, DNA Extraction, and Microarrays Analysis

Blood collection, DNA extraction, and DNA quality control were performed as recommended. DNA methylation analysis was performed according to Illumina Infinium HumanMethylation850 BeadChip EPIC array technology, as previously described [9]. In addition, blood cell heterogeneity analysis, bioinformatic pipeline, and statistical analysis were performed as described by Pragya et al. [9]. Briefly, blood samples were drawn into purple-top EDTA tubes at baseline and by the end of the 8 weeks of meditation sessions. A total of 34 pairs of samples were included, and the blood specimens were shipped to the Biorepository at the John P. Hussman Institute for Human Genomics at the University of Miami Miller School of Medicine. DNA extraction was performed according to the manufacturer’s recommendations on the Autogen FlexStar instrument (Holliston, MA, USA) (Catalog # AGKT-WB-640). After extraction, DNA quality control (QC) was performed using gel electrophoresis on a 0.8% agarose gel, and the DNA concentration was assessed using the Qubit dsDNA broad-range (BR) assay (ThermoFisher Scientific, San Diego, CA, USA).

### 2.5. Genome-Wide DNA Methylation and Quality Control and Downstream Analysis

Methylation arrays were conducted at the John P. Hussman Institute for Human Genomics Center for Genome Technology using validated protocols and fully automated liquid handling instrumentation (PerkinElmer, Shelton, CT, USA). DNA concentrations were normalized for the Illumina Infinium HumanMethylation850 BeadChip EPIC array. The samples were bisulfite converted according to Illumina specifications. The DNA input for bisulfite conversion was 500 ng. DNA samples were randomized across all arrays and scanned on an Illumina iScan (Illumina, San Diego, CA, USA). Raw.idat files were loaded into Illumina’s Genome Studio V1.0.0 software (Illumina, San Diego, CA, USA) for initial quality control and also were processed in R software (v4.1.1) with the minfi package for quality control [20]. All samples passed quality control criteria of (1) log median intensity in both the methylated and unmethylated channels over 10.5; (2) mean detection *p*-value across all probes less than 0.01. Individual probes were removed if the detection *p*-value was over 0.01 in any sample, if they were located on the sex chromosomes, if they contained a single nucleotide polymorphism with minor allele frequency ≥0.01 in the last five base pairs of the probe, or if they mapped to multiple positions in the genome [21]. To take into account the heterogeneity of cell type proportions across the samples, the proportion of 6 cell types (CD8 T cells, CD4 T cells, Natural Killer cells, B cells, monocytes, and Neutrophils) was estimated using the estimateCellCounts2 function in FlowSorted.Blood.EPIC R package [22,23]. The probes used in the estimation were removed from the subsequent analysis. Following quality control, data normalization was carried out in two steps. First, quantile normalization was applied using the lumiN function in the Lumi R package [24]. Second, beta-Mixture Quantile (BMIQ) normalization was applied to correct for the bias of type-2 probe values [25]. To remove potential batch effects, the ComBat method from the sva R package was applied [26,27]. Differential methylation analysis was performed broadly according to a recently published protocol for the analysis of methylation data primarily using the limma software package (Bioconductor version 3.17) with methylation M-value, log2 ratio of the intensities of methylated probe versus unmethylated probe, as the outcome. For each probe, a linear model was fitted (M-value ~ pre_treat + post_treat + CD8T + NK + Bcell + Mono + Neu), and empirical Bayes moderated t-statistics and *p*-values were generated [28,29]. To account for the samples from the sample subject, intra-subject correlations were estimated using a duplicate Correlation function in limma by including the subject ID as a blocking variable [30]. Probes with a nominal *p*-value below 0.05 and at least a 3% difference in methylation level were considered to be differentially methylated.

### 2.6. Correlation Study

The data analysis was carried out in the Julia computational Language framework (https://julialang.org, accessed on 5 June 2023). The modules employed are listed in the Appendix A, which contain all the functions discussed hereafter. To acquire insight on the potential relationships between the 470 methylated sites identified as differentially significant out of the ~850k total sampled sites (DMS) and the cognitive skill metrics, two separate correlation analyses were conducted. (1) The correlation between methylation changes in each of the 470 methylated sites and assessed cognitive skill, and (2) the correlations in methylation changes occurring among all 470 DMS. A Pearson correlation coefficient $r_{x,y}$ was used as a measure of linear correlation (or linear dependency) between dataset pairs (Equation (1)). The coefficient value may vary between $-1<r_{x,y}<1$. A value of $r_{x,y}=-1$ indicates a perfect negative correlation (for an increasing *x*, *y* decreases); conversely, a value of $\;r_{x,y}=1$ indicates a positive correlation (for an increasing *x*, *y* increases). However, if $r_{x,y}=0$, the two data are said to be uncorrelated. Values in the range [−1;1] indicate the degree of correlation.
(1)$$r_{x,y}=\frac{cov\left(x,y\right)}{\sigma_{x}\sigma_{y}}=\frac{\sum_{i=1}^{n}\left(x_{i}-\bar{x}\right)\left(y_{i}-\bar{y}\right)}{\sqrt{\sum_{i=1}^{n}{\left(x_{i}-\bar{x}\right)}^{2}\sum_{i=1}^{n}{\left(y_{i}-\bar{y}\right)}^{2}}}$$

To display the correlation calculation for the two analyses, a set of heatmaps will provide a qualitative visual tool to elucidate outcomes.

### 2.7. Further Statistical Analysis

Once the correlation matrices are generated, it is possible to further filter the dataset by finding and differentiating sites that display good and poor correlation and sites that appear to be recurrently positively or negatively correlated across tests. To this end, the datasets are first tested for normality, and then, a Student’s t-distribution test is carried out to determine if the correlation coefficient for each pair is significantly removed from an uncorrelated outcome. Normality was assessed using a Shapiro–Wilk test, which determines how closely a dataset approximates a normal distribution. The null hypothesis (H_o_) implies the sampled dataset is normally distributed p>0.05 (i.e., no significant difference from a standard normal distribution). If p<0.05, the null hypothesis is rejected, and the data are not normally distributed. For a sample size below 5000, the Shapiro–Wilk test’s *p*-value can accurately be estimated, and the outcome can be safely interpreted. The Student’s t-distribution test with n-2 degrees of freedom verifies whether the computed correlation coefficients are significant; in other words, with this test, we determine if each correlation coefficient obtained from paring samples is significantly removed from a hypothetical coefficient of 0 where no correlation is found (i.e., $r_{o}=0$).
(2a)$$t=\frac{r}{\sigma_{r}}=\frac{r_{xy}-r_{o}}{\sqrt{\frac{1-r_{xy}{}^{2}}{n-2}}}$$
(2b)$$\left\{\begin{array}{c} H_{o}:\;r_{xy}=r_{o},\;null\;hypothesis\;accepted\\ H_{1}:\;r_{xy}\neq r_{o},\;null\;hypothesis\;rejected \end{array}\right.$$
n represents the number of participants (in this case, n=34). Cognitive skill assessments were analyzed as baseline and 8 weeks postintervention comparison, using GraphPad prism version 8 (GraphPad software, Boston, MA, USA); a p value <0.05 was considered significant. 

## 3. Results

### 3.1. Cognitive Skills Assessments of the Different Meditation Groups

Following a set of paired *t*-tests with Bonferroni correction, the comparison between baseline and 8 weeks postintervention revealed statistically significant differences in four of the performance assessments. Interestingly, though, improvements were observed on nine measures. Briefly, the buzzing average, buzzing maximum, spatial recall, spatial recall processing, and positivity did not show significant changes in the 34 participants (p>0.01). Improvements were significant at the level of digital recall, listening recall, listening recall processing, and negativity (p<0.01; see Figure 1). Individual scores for all the participants are included in the Appendix A. Again, DMS were identified by selecting sites that showed a significant methylation differential of greater than 3% (p<0.05). This selection yielded a total of 470 DMS meeting these criteria after Bonferroni correction (reported in the Appendix A). The total 470 DMS and their genomic localizations, chromosome assignment, and potential functions can be found in the Appendix A.

### 3.2. Correlation between the Cognitive Skills and the 470 Differentially Methylated Sites

To correlate the cognitive skills and the methylation differential across all participants (*n* = 34), we (1) subtracted baseline scores from 8 weeks postintervention scores for every participant and (2) subtracted baseline methylation levels from 8 weeks post-intervention methylation levels for every participant (Equation (3)). Based on a significance level of 0.05 and sample size *n* = 34, a critical correlation value of ±0.34 was found (Equation (4)). We identified the 10 most positively and 10 most negatively correlated DMS with every cognitive skill. The data summarizing the correlation coefficient, p-value, DMS, genomic position, and chromosome localization are illustrated in the Appendix A. Table 1 depicts the most positively and negatively correlated DMS with every cognitive skill. In addition, Figure 2 shows the correlation between DMS and the nine cognitive skills.

The initial raw datasets for the overall methylation sites (~850k), reduced set of differentially methylated sites (470), and cognitive skill test outcomes for 34 participants were first imported, processed, and reorganized, to then evaluate the baseline data $x_{b}$ and 8-weeks postintervention data $\;x_{i}$ differences (Equation (3)).
(3)$$\Delta =x_{i}-x_{b}$$
(4)$$r_{c}=\sqrt{\frac{\frac{t_{c}^{2}}{n-2}}{\frac{t_{c}^{2}}{n-2}+1}}$$

Once the statistically significant relative difference ∆ for both site methylation and cognitive skill score were computed and compiled into matrices, the two correlation analyses were carried out. This table only reports a small portion (~3.8%) of the available observations given the rather large number of sites considered (470). Extended tabulation for the 20 highest DMS-to-cognitive skill correlations is provided in the Appendix A. The data displayed in Table 1 show a significant degree of correlation between site methylation and every cognitive skill. It can be seen that positive correlation reached as high as 0.62 (for cg17095850 on chr11 position 35311522), while inverse correlation reached −0.56 (for cg23768860 on chr7 position 47472944). Both significant outcomes occurred for the Negative Affect assessment factor. Mean positive correlation across all tests listed in the table is of 0.50, while the average negative correlation is of −0.48, which is well above the critical correlation coefficient. These findings suggest that maintaining a regular meditation practice is associated with measurable alterations in methylome and cognitive health. These results require confirmation at a larger sample size and different clinical trials before reaching a final conclusion about efficacy and feasibility. 

Based on the extended tabulation for the 20 highest site–cognitive skill correlation, it is possible to further filter the dataset considering the frequency of each gene across all cognitive skill tests. Table 2 provides a condensed summary of the site frequency for a minimum count of 3; an extended version of this table for a minimum count of 2 is provided in the Appendix A. Among the six sites (~1% of the 470 original sites) with the highest frequency detected, only one site has a frequency of 4 (~44%), whereas the other five sites have a frequency of 3 (~33%). If the exclusion criterium is reduced to a minimum of two occurrences (~22%) across all cognitive skill tests, an additional 28 sites are identified for a total of 34 (~7% of the 470 original sites).

The provided p-value (Table 1) is calculated based on the assumption that the correlation evaluations follow a normal distribution. The Shapiro–Wilk test reveals that the correlation values for all 470 sites and each cognitive skill are normally distributed aside from the “Digit Recall” test (p>0.05). The “Digit Recall” test reports a p-value of 0.0003, with a W = 0.9869 indicating that there is evidence that this particular cognitive skill correlation value may not be normally distributed. Figure 1 is meant to support the calculation reported in Table 3. 

The histogram representation (with 100 bins) in Figure 2 shows a seemingly normal distribution of correlation coefficients for cognitive skills about near-zero means (0.0320172, 0.0403665, −0.0297508, 0.026424, 0.0506869, 0.025163, −0.0186902, −0.000654581, and −0.00665924). Figure 2 also exhibits a number of correlation values that are significantly removed from each skill’s mean, indicating that, as previously observed, there can be a high degree of correlation between methylation sites and cognitive skill changes following a meditative practice. These tail values would typically be labeled as outliers, but in this analysis, the interpretation is different. A high correlation value (0.5<r<1 and −1<r<−0.5) identifies significant matching between methylome up/downregulation and cognitive skill performance improvements; high correlation counts are observed in less than 0.43% sites for each skill test (“Negative” and “Spatial Recall” tests). This, however, does not mean that other sites do not correlate. In fact, Figure 3 helps highlight the relative abundance of well-correlated sites in the form of a boxplot where the outliers and whisker values (1st and 4th quartile) represent good correlation outcomes. If the criteria for good correlation counts is extended (0.25<r<1 and −1<r<0.25), observations occur at an appreciably higher frequency for each skill test of up to 17% of sites (“Digit Recall” test). The boxplot provides additional insight on the data distribution, where most methylation sites are poorly correlated to any particular cognitive skill and each distribution is approximately centered at r=0 (no linear correlation detected).

Given the large datasets for the computed correlation data, heatmaps were generated as a visual tool to better grasp the correlation distributions for (1) site-to-cognitive skill and (2) site-to-sites (Figure 4). Figure 4A supports the claim that there is a significant amount of sites that are well correlated to each skill test (dark and light slivers), once again suggesting that the proposed meditation regime is linked to positive outcomes for participants through modifications in the regulatory methylome, ultimately influencing higher-order cognitive functions like memory processes and emotional states. Similarly, Figure 4B displays the site-to-site correlation coefficient heatmap highlighting a significant amount of high positively and inversely correlated locations (light and dark spots, respectively). The plot is symmetrical about the light-colored diagonal, which pairs each site to itself (representing the correlation matrix). The correlation matrix represented in Figure 4B was also used to evaluate the p-value about a hypothetical $r_{o}=0$, which revealed that the null-hypothesis (site-to-site correlation being close to 0) can be rejected in ~37% of cases. This helps support the data displayed in the figure, highlighting a significant amount of positively and negatively correlated sites. Detailed information relative to the nature of site-to-site correlation as well as a comprehensive mapping of correlation relationships may help elucidate which pathways are being activated and how.

To show in detail the previous observations relative to site-to-site and cognitive skill-to-site, correlation plots can be generated for each of the nine columns of Table 1. Figure 5 shows a sample, detailed set of plots displaying (1) the high positive and inverse correlation of two sites (cg01704474 and cg03261565) to the “Buzz Average” skill test exemplified by the linear curve fits slop magnitudes and (2) the high inverse correlation between those two sites. Similar figures have been generated for the following columns in Table 1 and are reported in the Appendix A. The scatter plots in Figure 5 report the raw methylation regulation data for all 34 participants revealing in detail how well-correlated test score and sites can be. In addition, the histograms show a reasonably well-distributed set of measured observations for either test or site complemented by the related heatmaps.

## 4. Discussion

In this report, we revealed the correlation between the performance differentials in cognitive skill assessments and changes at DMS in the human DNA methylome of healthy participants. Although the buzzing average, buzzing max, and the spatial recall did not show statistically significant differences between baseline and 8 weeks postintervention (Figure 1), our data revealed that some DMS correlated with these three cognitive skills. Buzzing average and buzzing max both correlated positively with cg01704474, which is located at the 5′UTR (untranslated region) of RNH1 on chromosome 11. RNH1 is a gene involved in hematopoietic-specific translation in human [31]. Conversely, cg03261565 was found to be the most negatively correlated with both cognitive skills, and it was not adjacent of any relevant gene. Additionally, digital recall was positively correlating with cg19060557, which is located at 5′UTR of USP13 gene coding for Ubiquitin Specific Peptidase 13. The USP13 gene is involved in multiple functions such as autophagy, mitochondrial energy metabolism, and DNA damage response [32,33,34,35,36]. On the other hand, our data revealed that digital recall is negatively correlating with the level of methylation at cg13049398, which is also found at the 5′UTR of a transcriptional factor ZNF156. This gene encodes a transcriptional repressor involved in malignant transformation, especially breast cancer and cellular proliferation through other cellular targets [37,38,39,40]. 

Listening recall was found to be positively correlating with cg23140777 of the unknown gene target on the human genome, while the listening recall was negatively correlating with cg06938061. This site was found to be in proximity to the TCERG1L gene, which codes for a transcriptional elongation regulator that is a key regulator of human obesity [41]. Interestingly, this site was among the most hypomethylated in the study participants (level of methylation was −10% at 8 weeks postintervention compared to baseline; see Appendix A). The listening recall processing was positively correlating with cg12128316, while it was negatively correlating with cg23561053, which is located on the promoter region of the TTLL7 gene. The TTLL7 gene codes for Tubulin Tyrosine Ligase like 7, which is involved in MAP2-positive neurites [42,43]. TTL7 is involved in tubulin glutamylation, which is known to be the most abundant tubulin post-translational modification occurring in the adult human brain [44].

The decline in negative affect was positively correlating with cg17095850, which is located in proximity of the SLC1A2 gene. This gene codes for a solute carrier transporter, especially the excitatory amino acid transporter 2 (EAAT2) in the brain [45,46,47,48,49,50,51]. This negative affect factor was found to be negatively correlating with cg23768860, which is localized in proximity of the TNS3 gene (Tensin 3), which is involved in signal transduction and cancer [52,53,54]. For the positive affect, our data revealed that this cognitive factor is positively correlating with cg06148656, which is localized 1500 bp from the transcriptional start site of the FBXO38 gene. This gene is mediating PD-1 ubiquitination and antitumor regulation in humans [55,56,57,58]. The analysis of positive affect relationships in our cohort of participants revealed that this cognitive factor is negatively correlating with cg13566979 present at the 5′UTR of the TBC1D5 gene. This gene codes for a GTPase-activating protein (GAP), which is involved in cellular trafficking through the membrane [59,60,61,62]. Spatial recall was found to positively correlate with cg05990364, which is adjacent to the HMP19 gene. HMP19 codes for a pancreatic cancer suppressor, especially in ductal adenocarcinoma (PADC) [63]. The spatial recall correlated negatively with cg22717379, which is found on chromosome 4 and is not in proximity of any gene of interest. Finally, the spatial recall processing was found to positively correlate with cg03333699, which is localized in chromosome 7 and was in proximity to the ADAP1 gene. ADAP1 also encodes for a GTPase-activating protein involved in cancer progression and brain memory function [64,65,66,67,68]. The spatial recall processing was found to be negatively correlating with the methylation level at cg00730266, which is at TSS1500 from the PPP1R9A gene at chromosome 7. This gene encodes for Neurabin1, which is a key candidate in protein in synaptic formation and function [69,70,71,72,73]. 

Our data revealed that several sites of hypo- and hypermethylated patterns were found to be correlated with improvements in performance on cognitive skills assessments in our participant cohort (healthy and novice college students). We presented the top 10 sites that are either positively or negatively correlating with every cognitive skill improvement (Appendix A). In our analyses, changes in additional sites involved in brain function, memory, synaptic trafficking, and other metabolic activities were identified, which potentially emphasize the urgent need to design more robust studies involving larger sample sizes of participants. Interestingly, some of the identified sites were found to be correlating with more than one cognitive skill, implying that Preksha meditation (PM) might have a signature of a cluster of genes that are common (Table 2). cg26094004 was found to correlate with four cognitive skills, and it is localized at the 5′UTR of the PYY gene on chromosome 17 and is involved in post-prandial appetite and glucose regulation [74,75]. cg03362824 was found to correlate with three cognitive skills and it was found to be at TSS200 from SKAP2, which is a gene coding for Src Kinase Associated Phosphoprotein 2 and is involved in inflammation suppression [76,77,78]. On the other hand, cg14862307 was close to a gene coding for STK4 which is a Serine/Threonine which in combination with STK3 (Serine/Threonine kinase 3) are key components in the Hippo Signaling pathway that is involved in cell proliferation and death [79]. Finally, the other sites which correlated with three cognitive skills were cg23632416, cg23191941, and cg12128316, and they were found in Open Sea on the genome.

Our sample size was 34 participants, which is a limiting factor in this study, but the exploration of similar studies with larger sample sizes will strengthen our understanding about the molecular elements correlating with cognitive skills improvements. Ultimately, we aim to use similar noninvasive and safe techniques to address diseases or syndromes that can be the next phase of future interventions in the medical field. We understand that our study is exploratory in nature, and the results are correlative. We believe that additional trials with larger sample size are required to confirm the results before adopting the meditation into routine practice in the clinical field.

## 5. Conclusions

In this study, we reported an association between the cognitive skills improvements and changes at the methylation levels on the human genome. In-depth insights are required to associate other outcomes such as clinical improvements measurements with similar changes at the genetic or epigenetic levels, especially using cost-effective, easy, and noninvasive intervention tools such as meditation. These interventions and others should be understood at the molecular levels before adopting them in the clinical field to treat diseases such as ADHD, autism, and other psychiatric diseases.

## Figures and Tables

**Figure 1 brainsci-13-01214-f001:**
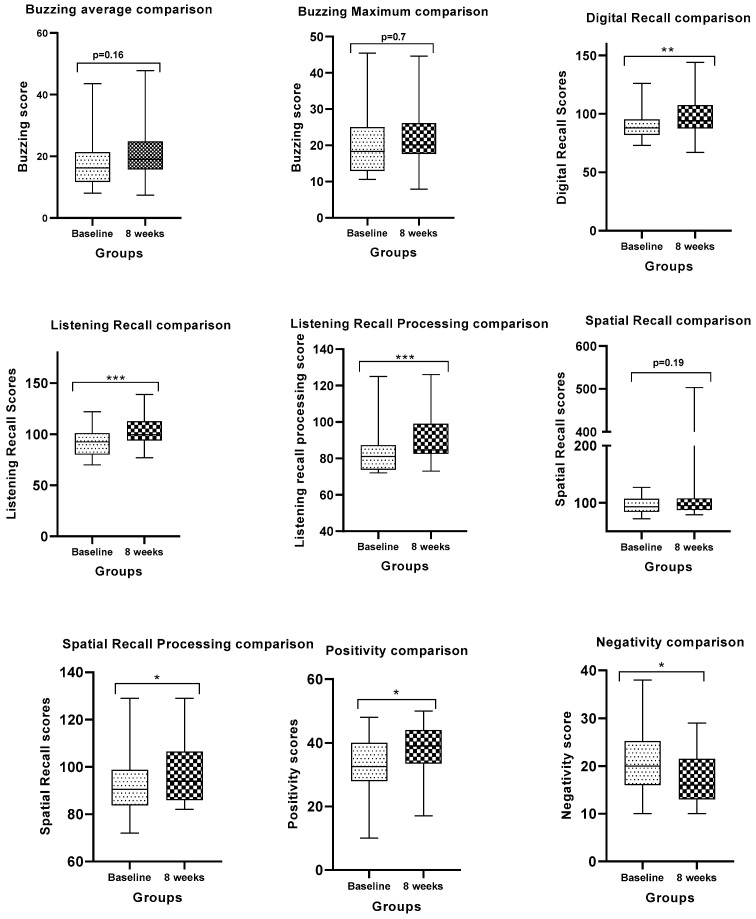
Comparison of the nine cognitive skills in the 34 participants baseline and 8 weeks postintervention. *p*-value < 0.05 was considered significant. *, **, and *** depicts *p*-value < 0.05, *p*-value < 0.001, and *p*-value < 0.0001, respectively.

**Figure 2 brainsci-13-01214-f002:**
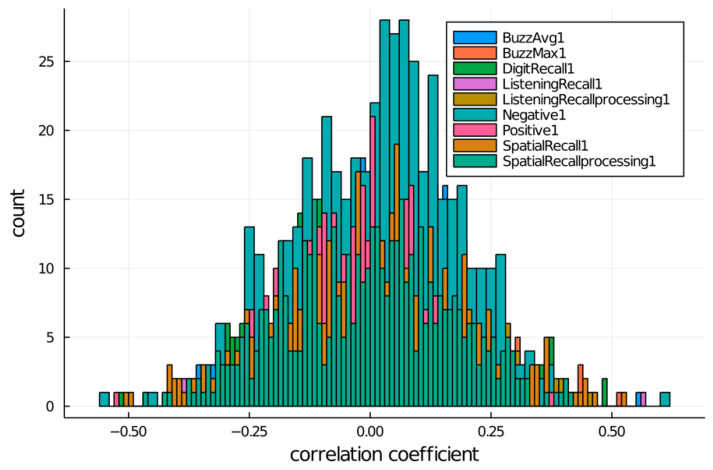
Histogram representation of correlation coefficient distribution for each cognitive skill test.

**Figure 3 brainsci-13-01214-f003:**
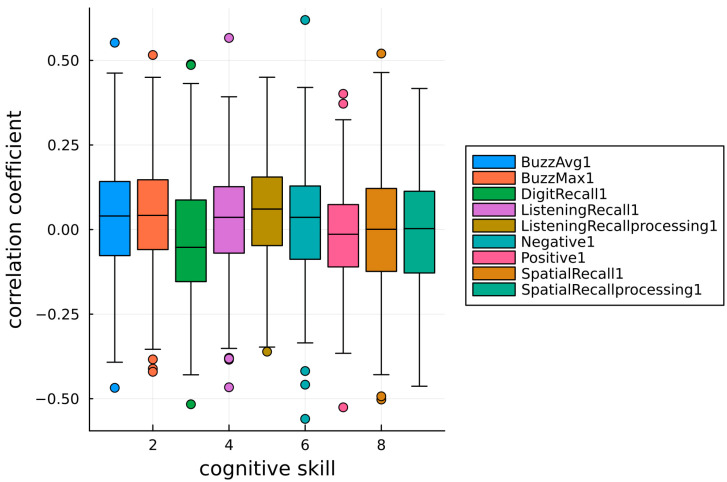
Boxplot representation of correlation coefficient distribution for each cognitive skill test.

**Figure 4 brainsci-13-01214-f004:**
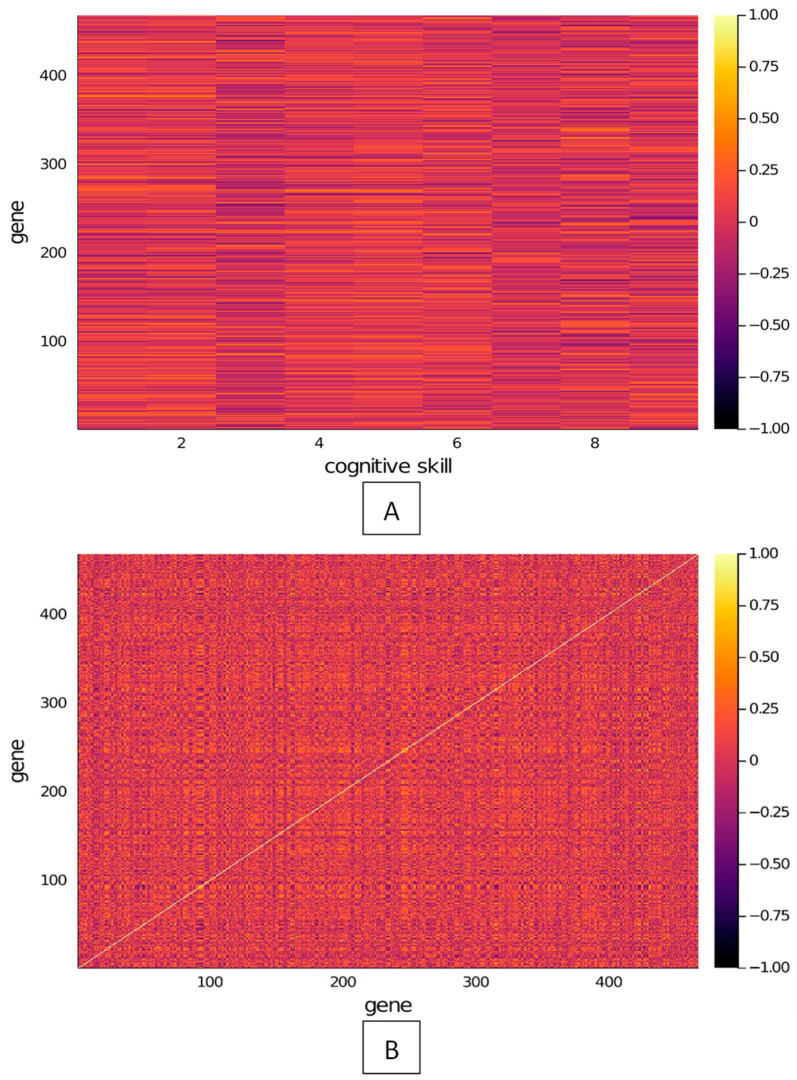
Pearson’s correlation coefficient heatmap for (**A**) site-to-cognitive skill and (**B**) site-to-site comparisons for all 9 cognitive performance assessments and all 470 differentially methylated sites identified in the study.

**Figure 5 brainsci-13-01214-f005:**
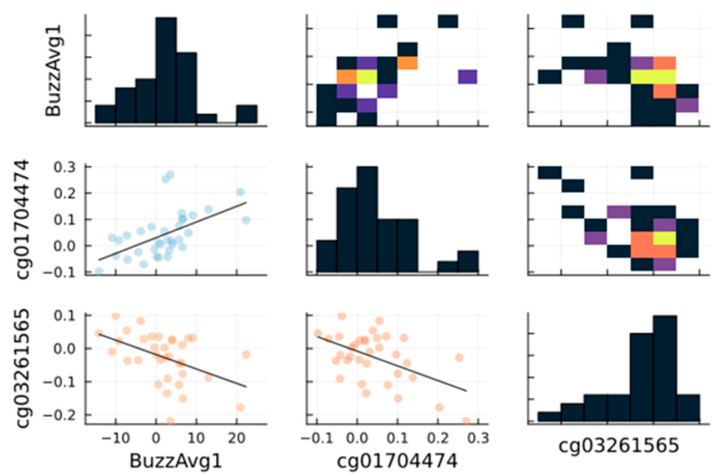
Correlation plot for Table 1 first column entries displaying the correlation between cg01704474 and the “Buzz Average” test, the correlation between cg03261565 and the “Buzz Average” test, and the correlation between cg01704474 and cg03261565. The scatter plot color code reflects the degree of correlation (blue = high positive correlation r>0, orange = high negative correlation r<0, and yellow = no correlation r=0). The histograms illustrate the distribution of each pair of correlated variables (e.g., skill test; sites). The heatmaps reflect the data distribution of the scatter plots (yellow = high density, black = low density).

**Table 1 brainsci-13-01214-t001:** Summary of correlation calculations between methylation sites and cognitive skill test reporting Pearson’s correlation r, its *p*-value relative to a hypothetical $r_{o}=0$, gene name, chromosomal localization, and positioning.

		Buzz Average	Buzz Max	Digit Recall	Listening Recall	Listening Recall Processing	Negative	Positive	Spatial Recall	Spatial Recall Processing
max positive	r	0.55	0.52	0.49	0.57	0.45	0.62	0.40	0.52	0.42
	** *p* ** **-value**	7.07 × 10^−4^	1.79 × 10^−3^	3.37 × 10^−3^	4.81 × 10^−4^	7.55 × 10^−3^	9.41 × 10^−5^	1.87 × 10^−2^	1.61 × 10^−3^	1.42 × 10^−2^
	**site**	cg01704474	cg01704474	cg19060557	cg23140777	cg12128316	cg17095850	cg06148656	cg05990364	cg03333699
	**Chromosome**	chr11	chr11	chr3	chr9	chr5	chr11	chr5	chr5	chr7
	**Position**	504918	504918	179399839	125457429	157137992	35311522	147808721	173493084	966569
max negative	r	−0.47	−0.42	−0.52	−0.47	−0.36	−0.56	−0.53	−0.50	−0.46
	** *p* ** **-value**	5.26 × 10^−3^	1.33 × 10^−2^	1.77 × 10^−3^	5.44 × 10^−3^	3.60 × 10^−2^	5.78 × 10^−4^	1.41 × 10^−3^	2.46 × 10^−3^	5.81 × 10^−3^
	**site**	cg03261565	cg03261565	cg13049398	cg06938601	cg23561053	cg23768860	cg13566979	cg22717379	cg00730266
	**Chromosome**	chr10	chr10	chr18	chr10	chr1	chr7	chr3	chr4	chr7
	**Position**	29312799	29312799	74157682	132942686	84465000	47472944	17712381	147939863	94537716

**Table 2 brainsci-13-01214-t002:** Condensed site frequency/count for top 20 site-cognitive skill test correlation, assuming a minimum occurrence of 3. An extended site frequency/count with a minimum occurrence of 2 is provided in the Appendix A.

CG Site	Buzz Average	Buzz Max	Digit Recall	Listening Recall	Listening Recall Processing	Negative	Positive	Spatial Recall	Spatial Recall Processing	Total Count
cg26094004	1	1	0	0	1	0	1	0	0	4
cg03362824	0	1	0	1	1	0	0	0	0	3
cg14862307	0	0	1	0	0	0	0	1	1	3
cg23632416	0	0	1	1	1	0	0	0	0	3
cg23191941	0	0	0	1	1	0	1	0	0	3
cg12128316	0	0	0	1	1	0	1	0	0	3

**Table 3 brainsci-13-01214-t003:** Shapiro–Wilk test outcome for each cognitive skill test reporting the W score, *p*-value, and interpretation. The null-hypothesis of this test assumes the data to be normally distributed; if *p* > 0.05, the null hypothesis is rejected.

Skill	W	*p*-Value	Normal
Buzz Average	0.9972	0.6258	true
Buzz Max	0.9989	0.9930	true
Digit Recall	0.9869	0.0003	false
Listening Recall	0.9957	0.2282	true
Listening Recall Processing	0.9952	0.1592	true
Negative	0.9958	0.2411	true
Positive	0.9977	0.7672	true
Spatial Recall	0.9983	0.9267	true
Spatial Recall Processing	0.9944	0.0850	true

## Data Availability

Data are available upon request.

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
