# Peer review of "Cognitive Skills and DNA Methylation Are Correlating in Healthy and Novice College Students Practicing Preksha Dhyāna Meditation"

_brainsci, 2023, doi:10.3390/brainsci13081214_

Round 1

Reviewer 1 Report

The paper presents an empirical study concluding that cognitive skills and DNA methylation are correlating in healthy and novice college students practicing meditation.

The research topic is relevant both for research and practice.

The manuscript is well written and understandable.

Important strengths include the practical research topic and the detailed assessments.

The paper could nicely fit into the Special Issue Complementary and Alternative Therapies for Mental Health.

The study is correlative. Causal claims need to be softened.

The sample of 34 participants is very small. Has an a priori power analysis been conducted? Are estimates reliable with such a small sample?

Preksha Dhyana meditation was used? The potential similarities and differences to other meditation procedures could be discussed in more detail.

What could be expected for other groups? E.g., older adults?

The conceptual contribution to theory advancement could be strengthened.

Practical implications could be detailed with more examples.

Author Response

Reviewer. The paper presents an empirical study concluding that cognitive skills and DNA methylation are correlating in healthy and novice college students practicing meditation. The research topic is relevant both for research and practice. The manuscript is well written and understandable. Important strengths include the practical research topic and the detailed assessments. The paper could nicely fit into the Special Issue Complementary and Alternative Therapies for Mental Health.

Comment: We thank the reviewer for his kind words about our work.

Reviewer: The study is correlative. Causal claims need to be softened.

Comment: We agree with the reviewer comments and our study went extra step to correlate the cognitive effects with differentially methylated sites. We are in the process to investigate these sites that are adjacent to certain genes using RT-qPCR to confirm the differential up and down regulation of such genes between baseline and 8 weeks post intervention.

 Reviewer: The sample of 34 participants is very small. Has an a priori power analysis been conducted? Are estimates reliable with such a small sample?

Comment: Once again the reviewer mentioned an interesting point about sample size. We agree on his opinion, and we mentioned this in the discussion as a limitation of the study. In addition, we consider our study as a pilot or feasibility study to plan future studies with larger sample size.

Reviewer: Preksha Dhyana meditation was used? The potential similarities and differences to other meditation procedures could be discussed in more detail.

Comment: We agree with the reviewer that there are several meditation protocols in the field. Several studies showed improvements in some clinical trials. We mentioned some examples in our manuscript, especially their effects on the gene expression profiling in humans.

 Reviewer: What could be expected for other groups? E.g., older adults?

 Comment: We are a group of PhDs and clinicians, and we are interested to apply Preksha meditation on IBS patients to relieve their symptoms such as abdominal pain with an emphasis to eventually reveal mechanistic pathways. We used our model to treat children with IBS and this work was published in Frontiers journal (2021). We added this study in the manuscript.

Reviewer: The conceptual contribution to theory advancement could be strengthened.

 Comment: We added more potential plans for meditation in the conclusion section.

Reviewer: Practical implications could be detailed with more examples.

Comment: Conclusion section was added to explain practical implication.

Reviewer 2 Report

This study aims to investigate the correlation between cognitive performance and DNA methylation. The goal is interesting itself. However, the analyses on the changes of cognitive performance and their association with changes in DNA methylation from 8-week Preksha Dhyana meditation are prone to false positive errors. I am not confident on the reported cognitive changes or associations between cognitive performance and DNA methylation because the authors did not correct for multiple comparisons (see my major concerns below). 

I have two major concerns related to the validity of these results.

1.    Pg. 4, line 169. You performed 9 comparisons and therefore it need multiple-comparison correction to claim whether the improvements are statistically significant after multiple comparison correction.

2.    Pg. 6, line 213-216. Again, you performed 470*9 = 4230 correlation models. so you need to perform multiple comparison corrections to only show the significant correlations after correcting for multiple comparisons.

A few other (mostly minor) concerns are listed below.

3.     Pg. 1, line 31-32. If improvements are not statistically significant, you cannot claim them as improvements. 

4.     Pg. 2, line 77-78. basic demographical information, such as age and gender, is needed here.

5.    Pg. 2, line 80-81. miss a verb in the sentence. Still need to mention how long each session was and how long participants practiced at home each week to understand the "dose" of meditation.

6.    Pg. 2, line 96. what is the range of variable time frame?

7.    Pg. 3, line 122. Not familiar with Julia Language. If it is a software, please give details, such as the link, company name, and etc.

8. Pg. 3, line 127-128. changes between baseline and follow-up? I am not clear whether you are evaluating the changes of genome and cognition or at baseline or at follow-up?

9. Pg. 3, line 145-146. Not sure if one-sample t test can be used to determine the significance of correlation coefficient?

10.  Pg. 4, line 151. where is 5000 coming from?

11.  Pg. 4, line 152-153. what is hypothetical mean and where it is coming from? for the mean of a dataset, what is a dataset? baseline data? follow-up data? or the change from the baseline and follow-up?

12.  Pg. 4, line 156. Not sure if the significance of a correlation coefficient is tested in this way (one-sample t test)?

13.  Pg. 4, line 159. what is the analysis here? correlation analysis?

14.  Pg. 5, Figure 1. state what * and ** and *** indicate in the figure caption.

15.  Pg. 5, line 182. What are scores here? you mean cognitive scores here?

16.  Pg. 5, line 189. Should be correlation distribution or correlation histogram.

17.  Pg. 6, line 198. what do you mean relative to 470 sites?

18.  Pg. 6, line 202. What do you mean two correlation analyses?

19.  Pg. 6, line 202. Direct is not the right terminology here. Also check other places using “direct”.

20.  Pg. 6, line 204-205. “magnitudinal removal” is not the right term.

21.  Pg. 12. References. Need more than one author for references. Please check the rule of the journal.

The authors need to work on some terminology related to statistics.

Author Response

Reviewer: This study aims to investigate the correlation between cognitive performance and DNA methylation. The goal is interesting itself. However, the analyses on the changes of cognitive performance and their association with changes in DNA methylation from 8-week Preksha Dhyana meditation are prone to false positive errors. I am not confident on the reported cognitive changes or associations between cognitive performance and DNA methylation because the authors did not correct for multiple comparisons (see my major concerns below). I have two major concerns related to the validity of these results.

Comments: We thanks the reviewer for the feedback, and we will try to answer his points below.

Reviewer: 1.    Pg. 4, line 169. You performed 9 comparisons and therefore it need multiple-comparison correction to claim whether the improvements are statistically significant after multiple comparison correction.

Comment: Thanks so much for feedback. We tested the nine cognitive skills as paired analysis for each cognitive skill (baseline vs 8 weeks postintervention). We used paired t test analysis, and we don’t think that a multiple comparison correction is needed as we did not test all of nine cognitive skills together. The individual scores for every participant in every cognitive skill measurement are provided in the supplementary data and it shows the scores as baseline (before meditation) and 8 weeks postintervention. We believe t test paired analysis will suffice this analysis unless the reviewer suggests an alternative test of choice.

Reviewer: Pg. 6, line 213-216. Again, you performed 470*9 = 4230 correlation models. So, you need to perform multiple comparison corrections to only show the significant correlations after correcting for multiple comparisons.

Comment: We appreciate the reviewer’s comments. In this study we performed a correlation analysis to address two separate inquiries. The primary interest is to determine if there exist a significant correlation between (direct or inverse) between methylation sites and “cognitive skills”, secondarily we want to determine if there exist any regulatory correlation between sites. As mentioned in the abstract (lines 33-35) and in lines 124-127 significant methylation sites were detected based on a 3% threshold and p-value<0.05.

A few other (mostly minor) concerns are listed below.

Reviewer: 3.     Pg. 1, line 31-32. If improvements are not statistically significant, you cannot claim them as improvements. 

Comment: Thanks for feedback. The reviewer is correct, the language relative to the changes in test performance between baseline and 8 weeks can be perceived as misleading, hence we have adjusted the language: Statistically significant improvements were observed in six of the nine assessments, predominantly relating to memory and affect.

Reviewer: 4.     Pg. 2, line 77-78. basic demographical information, such as age and gender, is needed here.

Comment: Thanks for the valuable point. Required information added with a description of Preksha meditation detailed protocol.

Reviewer: 5.    Pg. 2, line 80-81. miss a verb in the sentence. Still need to mention how long each session was and how long participants practiced at home each week to understand the "dose" of meditation.

Comment: fixed in the revised manuscript. Our meditation was guided by an expert trainer. We had strict adherence criteria as 80% of session attendance was required or the participant will be excluded. No meditation was done at home but rather under the guidance of certified Preksha trainer on site.

Reviewer: 6.    Pg. 2, line 96. what is the range of variable time frame?

Comment: the whole duration of Preksha was 8 weeks. We have blood samples withdrawn from the participants at baseline (before meditation), one hour after the first session and another one after 8 weeks. Due to cost issues, we compared baseline and 8 weeks.

Reviewer: 7.    Pg. 3, line 122. Not familiar with Julia Language. If it is a software, please give details, such as the link, company name, and etc.

Comment: We thank the reviewer for the feedback. Julia is an opensource coding language similar to python or R, hence we have included a URL-link to the main website when the language is first mentioned. The data analysis was carried out in the Julia computational Language framework (https://julialang.org).

Reviewer: 8. Pg. 3, line 127-128. changes between baseline and follow-up? I am not clear whether you are evaluating the changes of genome and cognition or at baseline or at follow-up?

Comment: We thank the reviewer for the opportunity to clarify the description. Before carrying out the two correlation studies described the alteration/changes in methylation (up/down regulation) at the 850K sites were evaluated by taking the difference in methylation levels between 8-weeks follow-up and baseline as expressed in equation 3. Similarly, changes in cognitive skill test performance were evaluated by taking the difference between test performance 8-weeks follow-up and baseline. Once the original 850k site were filtered with the state thresholds of +/-3% change and p-value<0.05, resulting in only 470 significant sites, we proceeded to estimate the correlation between (1) changes in cognitive skill tests and changes in methylation levels at each site and (2) changes in methylation levels at each site with respect to the changes in other sites.

Reviewer: 9. Pg. 3, line 145-146. Not sure if one-sample t test can be used to determine the significance of correlation coefficient?

Comment: The reviewer is correct the significant test for the correlation value takes a similar form to a one sample t-test but is adjusted to account for the correlation coefficient related standard deviation. Therefore, we have updated equation 2a to reflect this inaccuracy on our part and updated the p-value estimations in table in the manuscript and table S4 in the supplementary material. We have also clarified this in lines 151-154: The Student’s t-distribution test with n-2 degrees of freedom verifies whether the computed correlation coefficients are significant, in other words with this test we determine if each correlation coefficient obtained paring samples is significantly removed from a hypothetical coefficient of 0 where no correlation is found (i.e., ).

Reviewer:10.  Pg. 4, line 151. where is 5000 coming from?

Comment: We thank the reviewer for the opportunity to clarify. For sample size larger than 5000 the Shapiro-Wilk normality test outcome can provide inaccurate p-value estimation compromising the interpretation. In this study the sample size when testing for normality was well below this threshold therefore the reported normality test outcomes can safely be interpreted according to the known null hypothesis rejection criteria for (the null hypothesis being the sample being normally distributed, p-value<0.05 reject null hypothesis, p-value>0.05 accept null hypothesis). We have reworded the sentence in lines 149-151 to address any vagaries: For sample size below 5000, the Shapiro-Wilk test p-value can accurately be estimated, and the outcome can be safely interpreted.

Reviewer: 11.  Pg. 4, line 152-153. what is hypothetical mean and where it is coming from? for the mean of a dataset, what is a dataset? baseline data? follow-up data? or the change from the baseline and follow-up?

Comment: We appreciate the opportunity to clarify this passage. The initial raw data at baseline and 8-weeks “post-intervention” are used to calculate the change in methylation (up and down regulation) and in score performance by simply applying equation 3. The changes in methylation sites ( ) are first “filtered” using two criteria (1) a minimum change threshold of +/- 3% and a p-value<0.05. The computed difference  is then used to carry out the correlation studies to detect significant correlation (direct or inverse) between (1) the change in site methylation and change in cognitive skill performance and (2) the changes in a site with respect to other sites. Correlation significance was estimated with a Student’s t-distribution test with equation 2a which effectively determines whether a computed correlation coefficient is significantly removed from a coefficient of non-significant outcome (r=0). We have updated the passage in lines 151-154 to clarify.

Reviewer:12.  Pg. 4, line 156. Not sure if the significance of a correlation coefficient is tested in this way (one-sample t test)?

Comment: The reviewer is correct, when evaluating the correlation in a dataset a Student’s t-distribution test is used to determine significant according to equation 2a. We have addressed this in a short passage in lines 151-154: The Student’s t-distribution test with n-2 degrees of freedom verifies whether the computed correlation coefficients are significant, in other words with this test we determine if each correlation coefficient obtained paring samples is significantly removed from a hypothetical coefficient of 0 where no correlation is found (i.e., ).

Reviewer: 13.  Pg. 4, line 159. what is the analysis here? correlation analysis?

Comment: The analysis is a correlation analysis.

Reviewer: 14.  Pg. 5, Figure 1. state what * and ** and *** indicate in the figure caption.

Comment: Explained in the figure legend.

Reviewer: 15.  Pg. 5, line 182. What are scores here? you mean cognitive scores here?

Comment: Indeed, we mean the scores given to every participant at baseline and 8 weeks postintervention.

Reviewer: 16.  Pg. 5, line 189. Should be correlation distribution or correlation histogram.

Comment: We appreciate the chance to clarify the figure description. Following the first correlation analysis between methylation sites and cognitive skill performance, we wanted to depict the distribution of the calculated correlation coefficients grouped by cognitive skill test. Figures 2 and 3 offer great insight on the distribution and the existence of highly correlated pairs suggesting that further analysis may be warranted in future studies to investigate in detail these potential relationships. Therefore, we believe the proper description for figure 2 in particular ought to be correlation distribution.

Reviewer: 17.  Pg. 6, line 198. what do you mean relative to 470 sites?

Comment: We appreciate the reviewer’s feedback; the passage is indeed vague therefore we adjusted the sentence to avoid any misunderstanding as follows: The initial raw datasets for the overall methylation sites (~850k), reduced set of differentially methylated sites (470), and cognitive skill test outcomes for 34 participants were first imported, processed, and reorganized, to then evaluate the baseline data and 8-weeks post-intervention data  differences (Equation 3).

Reviewer:18.  Pg. 6, line 202. What do you mean two correlation analyses?

Comment: We appreciate the opportunity to clarify our analysis. As we outlined in lines 123-128 once we computed difference  for the change in cognitive skill performance and methylation site regulation the data is then used to carry out the correlation studies to detect significant correlation (direct or inverse) between (1) the change in site methylation and change in cognitive skill performance and (2) the changes in a site with respect to other sites. The two separate analyses provide fundamental outlooks on (1) the relationship between methylation sites and cognitive performance and (2) the relationships among methylation sites following intervention. These observations suggest that intervention induced detectable specific changes and provided a preliminary insight on what pathways may have been affected.

Reviewer: 19.  Pg. 6, line 202. Direct is not the right terminology here. Also check other places using “direct”.

Comment: We appreciate the reviewer’s comments. When characterizing the correlation coefficient using Pearson’s coefficient, the paired datasets are evaluated for linear correlation where the coefficient ranges between -1 and 1. Positive values indicate positive correlation (a linear curve with positive slope), while negative values indicate a negative correlation (a linear curve with negative slope). We have updated the terminology avoiding references to “direct” or “inverse” across the manuscript and the supplemental material.

Reviewer: 20.  Pg. 6, line 204-205. “magnitudinal removal” is not the right term.

Comment: The reviewer is correct, the terminology does not appropriately describe the statistical analysis carried out when determining the correlation significance, hence we have adjusted the passage in lines 201-204: Table 1 summarizes the correlation coefficient for the highest positively and negatively correlated sites for each cognitive skill, reporting the associated -value representing the correlation significance, gene name, chromosomal belonging, and chromosomal position.

Reviewer: 21.  Pg. 12. References. Need more than one author for references. Please check the rule of the journal.

Comment: The references were fixed upon the reviewer request. Thanks so much.

Reviewer 3 Report

This is an interesting study which investigates the correlation between cognitive skills and DNA Methylation after meditation intervention based on college students. This study can help us to understand more about the biological mechanism underlying the mediation intervention. Some issues need to be clarified and improved before publication:

1.       In the method part, although the authors’ previous work has been cited as references. It is not very clear for reviewers regarding the participants’ recruitment, intervention protocol, and DNA methylation bioinformatic analysis (especially for reviewers who are first time to read your work). Since those are important parts of your study design, please supplement those information again.

2.       Please clarify the test and retest reliabilities of two tests (“Conners Continuous Performance Test” and “Positive and Negative Affect Score”) with references.

3.       470 differentially methylated sites (DMS) were identified between the two time points (baseline and 8 weeks) without clarifying the multiple testing correction issue.

4.       For Figure 1, what’s the differences between *, **, and *** for the significant p-value. Please give an annotation.

5.       Please improve Figure 3 due to the content of the figure has been covered up by the annotation window.

6.       Line 80 has a typo: the “three” should be “were”.  

Author Response

Reviewer: This is an interesting study which investigates the correlation between cognitive skills and DNA Methylation after meditation intervention based on college students. This study can help us to understand more about the biological mechanism underlying the mediation intervention. Some issues need to be clarified and improved before publication.

Comment: We thanks the reviewer for valuable comments and feedback.

     Reviewer: 1.       In the method part, although the authors’ previous work has been cited as references. It is not very clear for reviewers regarding the participants’ recruitment, intervention protocol, and DNA methylation bioinformatic analysis (especially for reviewers who are first time to read your work). Since those are important parts of your study design, please supplement those information again.

     Comment: As per the reviewer request, we added the required information to the revised manuscript.

     Reviewer: 2.       Please clarify the test and retest reliabilities of two tests (“Conners Continuous Performance Test” and “Positive and Negative Affect Score”) with references.

    Comment: The tests were clarified, and reference were added to the revised manuscript.

   Reviewer:3.       470 differentially methylated sites (DMS) were identified between the two time points (baseline and 8 weeks) without clarifying the multiple testing correction issue.

   Comment: The comment of the reviewer is very interesting. In our analysis, we reported a nominal p value of 0,05 as a cutoff with 3% of methylation differences among all the 850000 sites on the chip. We can’t exclude the idea of having false positives among the 470 identified sites in this study. We are planning to select several up and down methylated sites to confirm the results in future report.

  Reviewer: 4.       For Figure 1, what’s the differences between *, **, and *** for the significant p-value. Please give an annotation.

 Comment: Added to the figure legend.

 Reviewer: 5.       Please improve Figure 3 due to the content of the figure has been covered up by the annotation window.

 Comment: The figure was replaced with a better legend.

 Reviewer: 6.       Line 80 has a typo: the “three” should be “were”.  

Comment: Fixed. We thank the reviewer for the feedback, the sentence clearly does not make much sense, the participants attended a total of 3 sessions per week. We have adjusted the sentence accordingly: Participants attended three guided PM sessions each week, for a total of eight weeks.

Round 2

Reviewer 2 Report

The authors did not address either of my major concerns for multiple comparison corrections. Either FWE or FDR corrections are needed in the study. You will get some significance by chance if you perform many statistical tests. They need to consult statisticians about this. 

N/A

Author Response

Reviewer: 1.    Pg. 4, line 169. You performed 9 comparisons and therefore it need multiple-comparison correction to claim whether the improvements are statistically significant after multiple comparison correction.

Comment: We appreciate the reviewer’s comments and the opportunity to clarify our analysis. In the initial assessment (before moving to the correlation analysis) of (1) changes in methylation levels (470 sites) and (2) changes in test performances (9 “cognitive skills”) a Bonferroni correction was applied to determine statistical significance following paired t-tests. We have realized that while we report the adjusted p-value in the supplemental material for the 470 methylation sites we have not reported the adjusted significance level for the performance test. In reference to the 9 comparison of the performance test statistical significance, we have now clarified this in the manuscript by reporting the adjusted significance (based on sequential Bonferroni) level thanks to reviewer’s feedback and updated related comments.

Reviewer: Pg. 6, line 213-216. Again, you performed 470*9 = 4230 correlation models. So, you need to perform multiple comparison corrections to only show the significant correlations after correcting for multiple comparisons.

Comment: We appreciate the reviewer’s comments. We took the opportunity to consult documentation and statisticians to determine if p-values associated to a correlation coefficient can or should be subject to correction for type-1 error. First, we would like to clarify that there is a distinction between comparison and correlation. When carrying out a comparison, the intention is to mainly understand how two population means differ (using a paired t-test for instance) and extending this to a group of paired comparison (i.e., considering more than one hypothesis u1≠u2, u1≠u3, u1≠u4, …) a correction becomes necessary, as the family wise error rate (FWER) increments as the group grows larger. A correlation on the other hand aims at elucidating a more complex relationship between two random variables, providing information relative to relative strength (0<|r|<1) and direction (r<1 or r>1). The related t-test that provides a p-value estimation for each correlation analysis (for a total of 470*9=4320) is in essence 470*9 one sample t-tests (Equation 2) that detect whether r is significantly different from ro=0, not other r values (i.e., a test to determine if a pair of data is significantly correlated). In this case the hypothesis is always the same (u1≠uo, u2≠uo, u3≠uo, …), which does not alter the false positives rate. If the computed 4320 were to be used to carry out a set of paired comparisons across the 9 performance tests to find if correlation detection differs significantly across tests, then we agree that a correction would be needed (this might be carried out in future studies following the reviewer’s suggestion). However, in this study we carried out exploratory statistics to determine (1) the correlation coefficients, (2) find/rank the largest positive/negative coefficients, and (3) determine the recurrence frequency (across performance tests) of highly correlated sites (table 2). The final step allowed us to detect methylation sites of potential interest for future studies, as such this analysis does not require multiple comparison correction as the reported p-values have not been used to determine significance in paired tests. We have taken the opportunity to calculate a critical correlation coefficient (rc, Equation 4) based on sample size (n=34 participants) and a significance level of 0.05. This resulted in rc=0.34, cleared by all table in table 1 and supplementary material S4. We hope this addresses the reviewer’s concerns and we are looking forward to any suggestion or feedback.